

# Effect of hypoxia conditioning on physical fitness in middle-aged and older adults—a systematic review and meta-analysis

Fanji Qiu[1], Jinfeng Li[2] and Liaoyan Gan[3]

[1] Movement Biomechanics, Institute of Sport Sciences, Humboldt-Universität zu Berlin, Humboldt Universität Berlin, Berlin, Germany
[2] Department of Kinesiology, Iowa State University, Ames, IA, United States of America
[3] Alberta International School of Recreation, Sport and Tourism, Beijing Sport University, Lingshui, Hainan, China

Corresponding author
Fanji Qiu,
fanji.qiu@student.hu-berlin.de

## ABSTRACT

**Background**. Hypoxic conditioning has emerged as a promising intervention for enhancing physiological adaptations. This systematic review and meta-analysis of randomized controlled trials aims to investigate the efficacy of hypoxic conditioning on physical fitness measures in aging populations.

**Methods**. The Embase, PubMed, Cochrane Library, and Web of Science were searched from inception to November 2024 (Prospero registration: CRD42023474570). The Cochrane Evaluation Tool and Grading of Recommendations Assessment, Development and Evaluation (GRADE) framework were used for risk of bias assessment and evidence certainty evaluation. Mean differences (MD) and standardized mean differences (SMD) and 95% confidence intervals (CI) were calculated using the Review Manager software. Subgroup analysis was performed to explore possible associations between the study characteristics and the effectiveness of the intervention.

**Results**. A total of 13 randomized controlled trials (RCTs) with 368 subjects were included in the meta-analysis. High certainty evidence found hypoxic conditioning (HC) significantly improved peak oxygen uptake ($VO_2peak$) (SMD = 0.31, 95% CI [0.01–0.61]; $P < 0.05$), while very low to moderate certainty evidence shown that hypoxic conditioning (HC) have not induced greater changes on functional outcomes (SMD = −0.21, 95% CI [−0.66–0.24]; $P > 0.05$), muscle strength (SMD = −0.19, 95% CI [−0.63–0.26]; $P > 0.05$), maximal power output (SMD = 0.29, 95% CI [−0.17–0.76]; $P > 0.05$), $VO_2max$ (SMD = −0.39, 95% CI [−1.12–1.90]; $P > 0.05$), and exercise workload (MD = −10.07, 95% CI [−34.95–14.80]; $P > 0.05$).

**Conclusion**. This study suggests that hypoxia conditioning has a greater effect on enhancing $VO_2peak$ compared to equivalent normoxic training in the middle-aged and older population. More high-quality RCTs are needed in the future to explore the optimal oxygen concentration and exercise intensity during hypoxia conditioning.

## INTRODUCTION

The global aging trend has led to a continuous increase in the middle-aged and older population (aged over 40 years) (*McNicoll, 2002*). Aging is a primary risk factor for chronic diseases, including cardiovascular disease, neurodegenerative conditions, which also impairs physical and cognitive functions, contributing to muscle atrophy (*Goodpaster et al., 2006*; *Niccoli & Partridge, 2012*; *Partridge, Deelen & Slagboom, 2018*). The loss of muscle strength, reduced oxygen uptake, and diminished physical function in the older population are closely associated (*Arnett et al., 2008*; *Gouveia et al., 2020*; *Sillanpää et al., 2014*), further reducing the quality of life. Thus, identifying strategies to address the decline in physical fitness among middle-aged and older adults is crucial.

Exercise has been proven to be an effective mean of improving physical fitness in middle-aged and older adults (*Valenzuela et al., 2023*; *Villareal et al., 2017*). Various forms of exercise, including aerobic exercise, resistance training, and high-intensity interval training (HIIT), have shown positive effects on the physical fitness of this population (*Hunter, McCarthy & Bamman, 2004*; *Marzuca-Nassr et al., 2020*; *Nash, 2012*). For instance, sedentary middle-aged and older adults who participated in a 16-week aerobic exercise intervention showed significant improvements in their 36-item short-form health survey (SF-36) scores compared to those who did not exercise (*Reid et al., 2010*). Additionally, healthy middle-aged and older individuals demonstrated significant enhancements in flexibility, stability, strength, and maximal oxygen uptake ($VO_2max$) after 12 weeks of HIIT (*Marzuca-Nassr et al., 2020*). The benefits of resistance training in enhancing muscle strength and function in middle-aged and older adults have also been well-documented (*Csapo & Alegre, 2016*; *Frontera et al., 1988*). However, despite the widely recognized benefits of exercise on physical fitness, approximately 25% of adults globally do not meet the recommended levels of physical activity (*WHO, 2021*). The percentage of older adults meeting the recommended weekly physical activity levels is even lower than that of younger populations (*Watson et al., 2016*), with only 2.5% to 22% of community-dwelling older adults achieving the World Health Organization-recommended levels of physical activity (PA) (150 min of moderate-intensity PA per week) (*Harris et al., 2009*; *Sagelv et al., 2019*). Therefore, more efficient methods are needed to help middle-aged and older adults manage their physical fitness.

Hypoxic conditioning (HC) is a non-pharmacological therapeutic intervention that induces systemic and/or localized tissue hypoxia by exposing individuals to low-oxygen conditions, either at rest or during exercise. Hypoxia elicits multisystemic physiological adaptations, including the upregulation of erythropoietin synthesis to stimulate erythropoiesis (*Ehrenreich et al., 2022*; *Rodríguez et al., 1999*), the potentiation of sympathetic nervous system activity with concomitant augmentation of ventilatory response (*Burtscher et al., 2024*) and stimulating the increase in mitochondrial electron transfer efficiency to maintain adenosine triphosphate (ATP) production (*Fukuda et al., 2007*). This approach can lead to similar or enhanced physiological and functional benefits compared to normoxic training (*Hobbins et al., 2017*; *Samuel & Franklin, 2008*; *Verges et al., 2015*). This technique is often used as a training strategy to enhance athletic performance

(*Millet et al., 2010*), and has also been shown to improve physical function in non-athletic individuals. Additionally, it can be safely and effectively used to manage elevated heart rate and blood pressure in older patients with cardiovascular diseases (*Glazachev et al., 2021*; *Maher, Jones & Hartley, 1974*; *Nishimura et al., 2010*). For example, eight weeks of hypoxic conditioning led to greater muscle hypertrophy in healthy young individuals compared to exercise alone (*Kurobe et al., 2015*). In the middle-aged and older population, eight weeks of hypoxic conditioning significantly improved peak oxygen consumption ($VO_2$peak) and maximal power output ($PO_{max}$) in obese participants (*Chacaroun et al., 2020*). A four-week intervention combining aerobic exercise with intermittent hypoxia significantly increased the time to exhaustion in older adults during an exercise load test (*Schega et al., 2016*). Importantly, hypoxic conditioning does not increase perceived exercise intensity and may even reduce the overall exercise workload (the blend of volume and the intensity in training) (*Girard, Malatesta & Millet, 2017*; *Scott et al., 2018*; *Wiesner et al., 2013*). The efficacy of hypoxic conditioning remains an ongoing scientific debate. Study have demonstrated that training under hypoxic conditions may surpass normoxic exercise in enhancing lean body mass and muscle fiber cross-sectional area. This purported superiority is attributed to the synergistic effects of mechanical loading and hypoxia-induced metabolic stress, which collectively enhance motor unit recruitment, stimulate anabolic hormone secretion, and promote myofibrillar protein synthesis (*Jung et al., 2021*). However, conflicting evidence suggests that hypoxic exposure may paradoxically accelerate skeletal muscle catabolism through impairment of protein synthesis mechanisms (*Chen et al., 2014*). Notably, this debate also exists among middle-aged and older people. For instance, after eight weeks of hypoxic endurance training, participants with an average age of 62 years showed no differences in constant workload test results or $VO_2$peak compared to those who trained under normoxic conditions (*Chobanyan-Jürgens et al., 2019*). Additionally, studies involving obese middle-aged and older participants found that five or 18 weeks of hypoxic conditioning did not result in significantly higher $VO_2$peak and $PO_{max}$ compared to equivalent training under normoxia (*Camacho-Cardenosa et al., 2020*; *Törpel, Peter & Schega, 2020*).

Therefore, this study aims to systematically review and conduct a meta-analysis to analyse the effects of training under hypoxic conditions in physical fitness outcomes compared to the same training under normoxic conditions in middle-aged and older adults. Adding hypoxia to exercise may further enhance physical fitness outcomes in this population.

## MATERIALS & METHODS

This systematic review and meta-analysis was conducted in accordance with the Preferred Reporting Items for Systematic Review and Meta-analyses (PRISMA) 2020 Statement (*Page et al., 2021*) and registered at PROSPERO (CRD42023474570). We determined the inclusion criteria based on the PICOS principles: (1) Participants, (2) Interventions, (3) Comparisons, (4) Outcomes, (5) Study Design.

## Eligibility criteria

### Participants

Studies were eligible if participants were middle-aged (40–65 years) and older adults (over 65 years). Excluded criteria: (1) participants aged under 40; (2) participants are suffering from/ suffered from severe health conditions, including heart disease, pulmonary fibrosis, and pulmonary arterial hypertension. There were no limitations based on sex, ethnicity, or involvement in sports.

### Interventions

Studies were eligible if the interventions used aerobic training, resistance training, whole-body vibration or other types of exercises which were combined with hypoxia. There were no restrictions on intervention duration and settings.

Exclusion criteria: in addition to exercise and hypoxic conditions, other interventions were also employed (such as the use of supplements like Rhodiola and oxygenated water). Applying hypoxia *via* local techniques, such as blood flow restriction.

### Comparisons

Studies were eligible if the control groups of included studies were subjected to same exercise protocol as intervention group, but trained under normoxic conditions.

Exclusion criteria: the control groups used the same exercise protocol as the intervention group, but underwent a different hypoxia protocol.

### Outcomes

Studies qualified for inclusion if they contained at least one result of physical fitness. The interested outcomes of physical fitness including functional outcome scores and results of exercise tests (*e.g.*, $VO_2max$, $VO_2peak$, repetition maximum (1RM)). We use leg extension strength as a proxy for muscle strength.

Exclusion criteria: studies that did not include results on physical fitness.

### Study design

Studies were included with criteria: (1) randomized controlled trials (RCTs); (2) include studies published in English only, as most high-quality RCTs were published in English (*Morrison et al., 2012*).

Exclusion criteria: (1) duplicate publications; (2) study protocol; (3) letters; (4) conference abstracts; (5) animal experiments; (6) literature review articles. Articles without full text or on which data extraction could not be performed were also excluded.

## Search strategy and study selection

PubMed, Cochrane, Embase, and Web of Science were searched for relevant studies published before December 2024. The search was limited to studies published in English and focused on keywords related to the population (*e.g.*, middle-aged or older adults), interventions (*e.g.*, hypoxia, exercise training), and outcomes related to physical fitness. In PubMed, the MeSH database in combination with MeSH terms and entry terms were applied, with adjustments made to refine the search strategies across the different databases. Additionally, filters applied included "randomized controlled trial (publication types),"
"randomized (title/abstract)," and "placebo (title/abstract)." The complete search strategy in four databases is presented in the File S1. All search outcomes were imported into Endnote X9 (Thomson Reuters, NY, USA), where they were collated and any duplicates were removed.

Additionally, manual searches were performed according to the references of the included articles. Two authors (FJQ, JFL) independently screened and reviewed the titles and abstracts of the identified studies. After excluding ineligible literature, they reviewed the full texts to finalize the selection of the studies. Any disagreements were resolved through consultation with the third author (LYG).

## Data extraction

FJQ and JFL conducted the data extraction and results compilation independently. The data extracted from articles were uploaded to Microsoft Excel. In cases of disagreement during the extraction process, a third researcher (LYG) was consulted for resolution. For three studies that did not provide original data on physical fitness outcomes, we used Graph Digitizer software (Digitizelt, Braunschweig, Germany) to extract data from the figures presented in the articles. The extracted data from the studies were: first author, year of publication, population characteristics (*e.g.*, age, male/female sex, body mass index (BMI)), sample sizes in study groups, intervention details: settings of hypoxia (*e.g.*, inspired oxygen fraction, duration), characteristics of exercise training (*e.g.*, exercise intensity, type of exercise), intervention duration (in weeks/months), population type (middle-aged or older adults), reported results (*e.g.*, outcomes of cardiorespiratory function, muscle strength). For data reported as "mean ± SE/SEM (standard error/standard error of the mean)," we used the following formula to calculate the standard deviation (SD):

$$SD = SE \times \sqrt{n}.$$

## Evaluation of risk of bias

Two reviewers (FJQ and JFL) independently evaluated the risk of bias of the included literatures according to the Cochrane Risk of Bias Assessment Tool guideline (Version 1.0) (*Higgins et al., 2011*). There were seven domains evaluated in this tool, including random sequence generation, concealment of allocation, blinding of participants, investigators, and assessors, blinding of outcome assessment, incomplete outcome data processed, selective reporting bias, other bias. Each domain was rated to indicate the level of bias: low (+), uncertain (?), or high risk (−). Any discrepancies generated during the assessment process were addressed through discussion or by consulting a third researcher for dispute resolution.

Given that each comparison group included fewer than 10 studies, funnel plots were not graphed to detect publication bias (*Sterne et al., 2011*). Sensitivity analyses were performed to explore the sources of heterogeneity (*Higgins et al., 2019*). The robustness of the pooled results was assessed using exclusion-by-exclusion sensitivity analysis, removing one study at a time from the pooled analysis, systematically excluding one study at a time to determine if the outcomes of the remaining studies remained consistent with those of the complete dataset.

## Quality of evidence

The Grading of Recommendations Assessment, Development and Evaluation (GRADE) system was used to evaluate the overall certainty of the evidence for each outcome (*Guyatt et al., 2008*). The certainty of the evidence was rated as high, moderate, low, or very low.

## Statistical analysis

Meta-analysis were conducted with Review Manager version 5.4 (The Cochrane Collaboration, Copenhagen, Denmark). The continuous variables were expressed as mean difference (MD) with 95% confidence intervals (CIs). The standardized mean differences (SMD) and 95% CIs were applied when the outcomes were reported not in the same unit (*Higgins et al., 2019*). A negative MD/ SMD favours the experimental group and a positive SMD favours the control group. The magnitude of heterogeneity 50% or a *p*-value less than or equal to 0.10 in the *Q* test indicates the presence of moderate to high heterogeneity. Pooled effects were calculated using random-effects and fixed-effects models according to the magnitude of heterogeneity, taking into account the variations across studies and assigning appropriate weights to each study. To investigate potential associations between study characteristics and the effectiveness of intervention and investigate potential sources of heterogeneity, subgroup analyses were conducted, including age (middle-aged or older), obese (yes or not), duration of hypoxia conditioning (≤8 weeks or >8 weeks), inspired oxygen fraction ($FiO_2 \leq 15\%$ or $FiO_2 > 15\%$), hypoxia duration per session (≥60 min or <60 min), and intervention type (aerobic or resistance training). Statistical significance was set with *p* value of 0.05.

# RESULTS

## Search yield

During the study selection phase, a total of 215 articles were identified as potentially eligible (PubMed: $n = 18$, Embase: $n = 41$, Web of Science: $n = 128$, Cochrane: $n = 26$; manual search: 2). Following the exclusion of duplicates, 186 articles remained for screening. A total of 13 articles (*Allsopp et al., 2022*; *Camacho-Cardenosa et al., 2020*; *Chacaroun et al., 2020*; *Chobanyan-Jürgens et al., 2019*; *Gatterer et al., 2015*; *Ghaith et al., 2022*; *Klug et al., 2018*; *Park et al., 2019*; *Schega et al., 2016*; *Schega et al., 2013*; *Schreuder et al., 2014*; *Törpel, Peter & Schega, 2020*; *Wiesner et al., 2013*) were finally included in the meta-analysis after screening full-texts. This meta-analysis assessed an experimental group with 188 participants and a control group with 180 participants (Fig. 1).

## Characteristics of included studies

The included RCTs utilized hypoxic conditioning in the experimental groups, with the control groups receiving only exercise. All studies reported on the variables of interest. To summarize, the 13 qualified studies (Table 1) included 368 participants (female 44%), with age ranging from 42.1 (1.7) to 72.6 (3.57) years. Across these studies, interventions combined hypoxic conditions with exercise, compared against exercise alone in normoxic environments. All the studies were performed under normobaric conditions. The interventions included aerobic exercise ($n = 7$), HIIT ($n = 1$), resistance

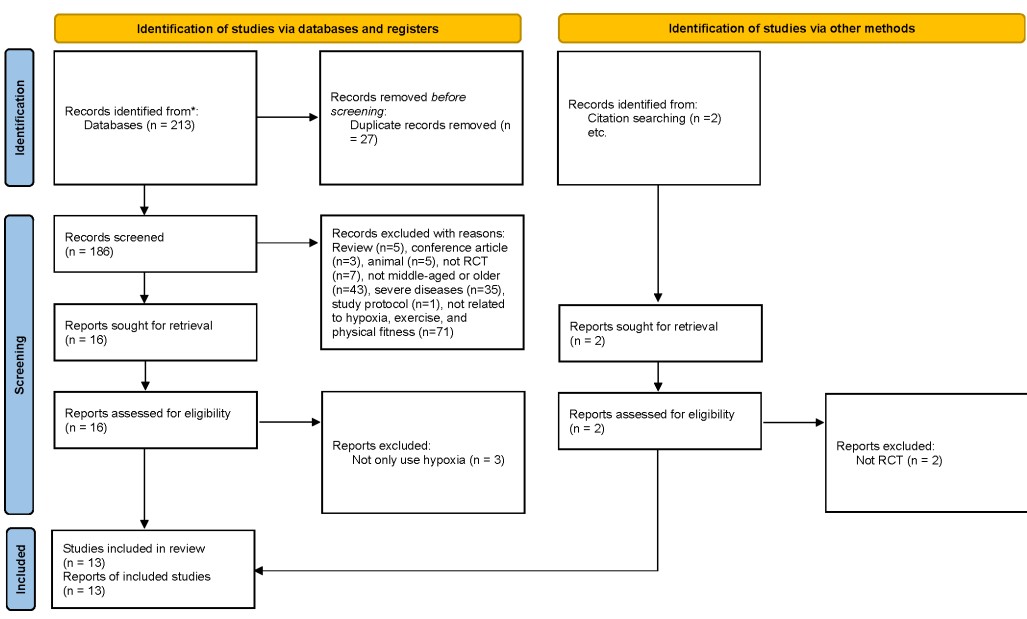

**Figure 1  Study selection process (according to the PRISMA guidelines).**

training ($n = 3$), whole-body vibration ($n = 1$), and aerobic exercise combined with resistance training ($n = 1$). Three of the included studies assessed the functional outcomes (*Camacho-Cardenosa et al., 2020*; *Park et al., 2019*; *Schega et al., 2013*), three of the included studies examined the muscle strength (*Allsopp et al., 2022*; *Park et al., 2019*; *Törpel, Peter & Schega, 2020*), seven studies examined power output (*Chacaroun et al., 2020*; *Chobanyan-Jürgens et al., 2019*; *Gatterer et al., 2015*; *Klug et al., 2018*; *Schega et al., 2016*; *Schreuder et al., 2014*; *Törpel, Peter & Schega, 2020*), five studies reported VO$_2$max (*Allsopp et al., 2022*; *Chacaroun et al., 2020*; *Ghaith et al., 2022*; *Klug et al., 2018*; *Wiesner et al., 2013*), and six studies examined VO$_2$peak (*Chacaroun et al., 2020*; *Chobanyan-Jürgens et al., 2019*; *Gatterer et al., 2015*; *Ghaith et al., 2022*; *Schreuder et al., 2014*; *Törpel, Peter & Schega, 2020*). The intervention periods across the thirteen articles ranged from 4 weeks to 8 months, with hypoxia exposure durations ranging from 26 to 180 min.

## Results of risk of bias evaluation

Detailed information on the risk of bias in randomized controlled trials can be found in Fig. 2. We found that out of 13 articles, 11 did not clearly specify how they generated random sequence and conducted allocation concealment, leading to an unclear risk of selection bias. Approximately half of the studies exhibited performance bias and attrition bias. Only three studies were assessed with low risk of bias in blinding of outcomes assessment and three with high risk of bias in incomplete outcome data.

## Meta-analysis
### Results of functional outcomes analysis

Three experiments involving 77 participants reported functional outcomes using the 36-Item Short Form Health Survey - physical component, time to exhaustion, and timed "Up

**Table 1  Characteristics of included studies.**

| Study | Study group | Sample size (M/F) | Age | BMI | Duration | Frequency | Intervention type | Exercise intensity | FiO$_2$/ duration per session | Outcome |
|---|---|---|---|---|---|---|---|---|---|---|
| *Allsopp et al. (2022)* | H | 10 (6/4) | 65.9 ± 1.1 | 24.9 (1.1) | 8wks | 2d/wk | RT | 70% 1RM | 14.4%/60 min | VO$_{2max}$, LE 5RM |
| | N | 10 (6/4) | 64.0 ± 0.8 | 23.9 (0.8) | | | | | | |
| *Camacho-Cardenosa et al. (2020)* | H | 9 (2/7) | 72.56 (3.57) | 28.31 (4.39) | 18wks | 2d/wk | WBV | 12.6 Hz/4 mm | 16.1%; 26min | TUG |
| | N | 10 (5/5) | 68.80 (5.33) | 27.76 (5.80) | | | | | | |
| *Chacaroun et al. (2020)* | H | 12 (11/1) | 52.0 (12.0) | 31.2 (2.4) | 8wks | 3d/wk | Aerobic | 75% HR$_{max}$ | ∼13.0%/45 min | VO$_{2peak/ max}$, PO$_{max}$ |
| | N | 11 (8/3) | 56.0 (11.0) | 31.8 (3.2) | | | | | | |
| *Chobanyan-Jürgens et al. (2019)* | H | 14 (7/7) | 60.4 (5.1) | 28.6 (3.0) | 8wks | 3d/wk | Aerobic | 60%–70% VO$_{2peak}$ | 15%; 30–40 min | VO$_{2peak}$, PO$_{max}$, WL |
| | N | 15 (7/8) | 63.8 (5.8) | 28.3 (1.9) | | | | | | |
| *Gatterer et al. (2015)* | H | 16 (4/12) | 50.3 (10.3) | 37.9 (8.1) | 8 m | 2d/wk | Aerobic | 65–70% HR$_{max}$ | 12.2%–14.0%/180 min | VO$_{2peak}$; PO$_{peak}$ |
| | N | 16 (6/10) | 52.4 (7.9) | 36.3 (4.0) | | | | | | |
| *Ghaith et al. (2022)* | H | 16 (10/6) | 51.0 (8.3) | 31.5 (4.0) | 8wks | 3d/wk | HIIT | 80–100% workload$_{max}$ | ∼12%/60min | VO$_{2max/peak}$; PO$_{max}$; WL |
| | N | 15 (13/2) | 52.0 (7.5) | 32.4 (4.8) | | | | | | |
| *Klug et al. (2018)* | H | 12 (12/0) | 55.0 (2.1) | 35.5 (1.4) | 6wks | 3d/wk | Aerobic | 50–60% HR$_{max}$ | 15.0%/60min | VO$_{2max}$, WL |
| | N | 11 (11/0) | 57.6 (2.2) | 34.1 (0.9) | | | | | | |
| *Park et al. (2019)* | H | 12 (12/0) | 66.50 (0.90) | 26.00 (0.61) | 12wks | 3d/wk | Aerobic; RT | 60–70% HR$_{max}$ | 14.5%/90-100 min | TUG, GS |
| | N | 12 (12/0) | 66.50 (0.67) | 25.63 (0.35) | | | | | | |
| *Schega et al. (2013)* | H | 17 (4/13) | 63.7 (3.4) | 27.34 (5.0) | 6wks | 3d/wk | RT | 50% $F_{max}$ | SpO$_2$ = 90%–80%/60min | SF-12 |
| | N | 17 (4/13) | 63.6 (3.2) | 27.58 (4.3) | | | | | | |
| *Schega et al. (2016)* | H | 17 (9/9) | 66.4 (3.3) | - | 4wks | 3d/wk | Aerobic | 65%–75% HR$_{max}$ | SpO2 = 90%–80%/90min | TTE, PAP |
| | N | 16 (10/8) | 67.9 (4.4) | | | | | | | |
| *Schreuder et al. (2014)* | H | 10 (9/1) | 57 (6) | 30.9 ± 4.1 | 8wks | 3d/wk | Aerobic | 70%–75% HRR | 16.5%/45min | VO$_{2peak}$, WL |
| | N | 9 (5/4) | 52 (8) | 36.0 ± 6.5 | | | | | | |
| *Törpel, Peter & Schega (2020)* | H | 19 (9/10) | 68.1 (4.6) | 27.6 (4.2) | 5wks | 4d/wk | RT | 40% 1RM | SpO$_2$= ∼80–85%/180 min | VO$_{2peak}$; $F_{max}$KE; PO$_{max}$ |
| | N | 17 (9/8) | 67.8 (4.1) | 26.9 (3.6) | | | | | | |
| *Wiesner et al. (2013)* | H | 24 (10/14) | 42.2 (1.2) | 33.1 (0.3) | 4wks | 3d/wk | Aerobic | 65% VO$_{2max}$ | 15.0%/60min | VO$_{2max}$ |
| | N | 21 (8/13) | 42.1 (1.7) | 32.5 (0.8) | | | | | | |

**Notes.**

FiO$_2$, inspired oxygen fraction; $F_{max}$, maximum force; $F_{max}$KE, $F_{max}$ of knee extension; GS, Grip strength; H, hypoxia; HRR, heart rate reserve; Hz, hertz; LE 5RM, Leg extension 5RM; N, normoxia; PAP, peak aerobic power; PO$_{peak}$, peak power output; RT, resistance training; SF-12, short-Form 36-Item health survey - physical component; SpO2, oxygen saturation of the blood; TTE, time to exhaustion; TUG, Timed "Up and Go"; VO$_{2peak}$, peak oxygen uptake; VO$_{2max}$, maximal oxygen uptake; WBV, Whole-body vibration; WL, workload.

(a)

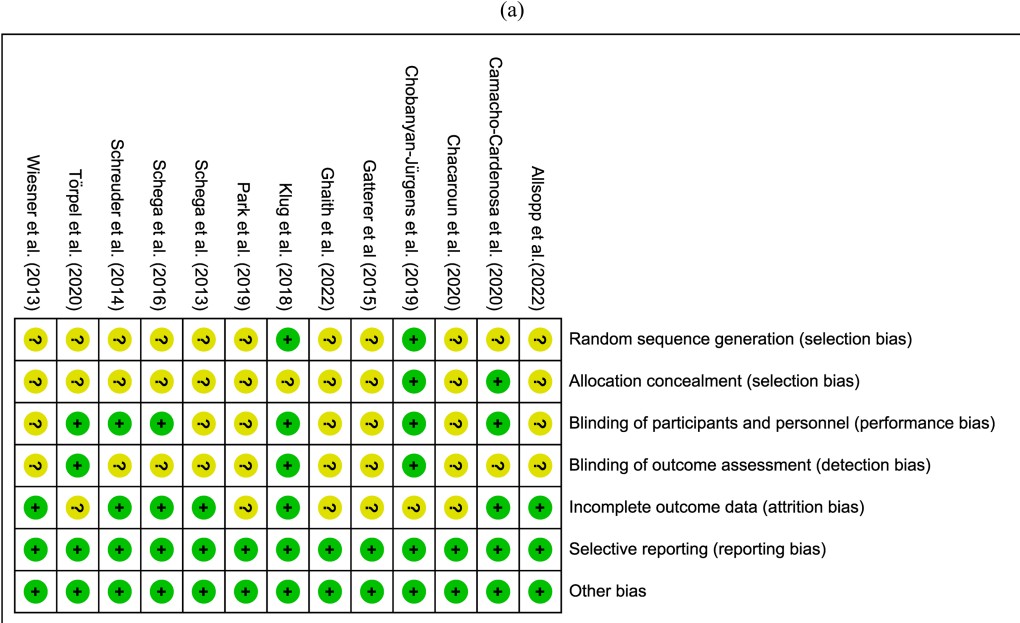

(b)

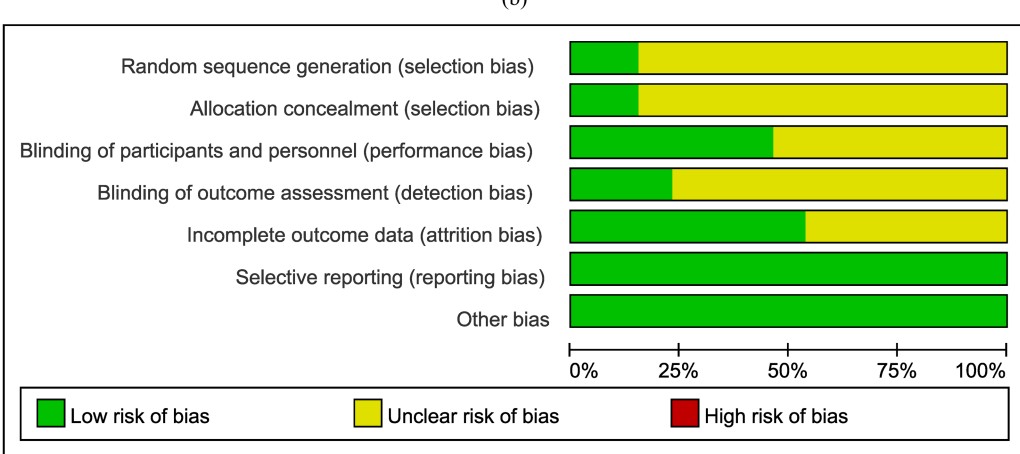

Figure 2 **Risk of bias assessment.** Where the ''green'' represents low risk of bias; ''yellow'' represents un-clear risk of bias; ''red'' represents high risk of bias.

and Go'' test. Moderate certainty evidence (Fig. 3A) revealed that the combined strategy had the similar effect on functional outcomes compared to exercise alone (SMD = −0.21, 95% CI [−0.66–0.24]; low heterogeneity), the certainty of evidence was downgraded due to differences in outcome measures. Subgroup analysis (Table 2) did not reveal significant effects based on age, obesity, intervention duration, hypoxia duration, inspired oxygen fraction, and intervention type. Sensitivity analysis did not reveal any differences in the pooled results after removing a single study.

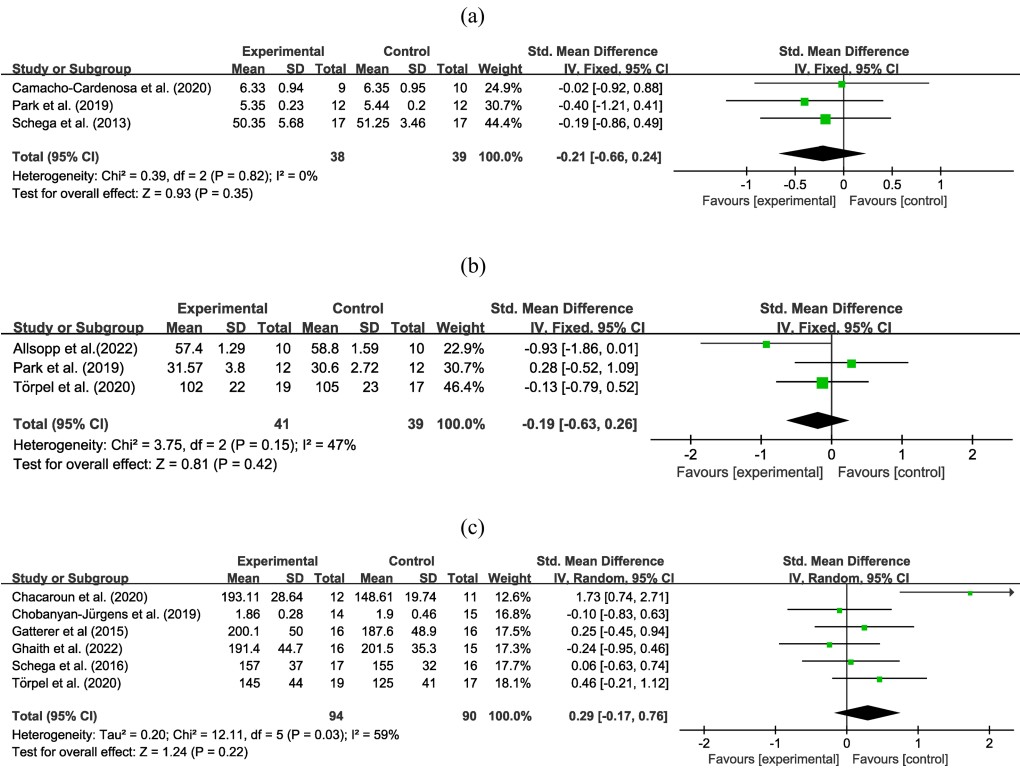

**Figure 3** Forest plot of the meta-analysis on (A) functional outcomes, (B) muscle strength, and (C) maximal power output.

## Results of muscle strength analysis

Three trials with 80 participants reported an outcome on muscle strength. Moderate certainty evidence (Fig. 3B) revealed that combined strategy had no superior improvement on muscle strength than exercise alone (SMD = −0.19, 95% CI [−0.63–0.26]; low heterogeneity), the certainty of evidence was downgraded due to high heterogeneity. Subgroup analysis (Table 2) did not reveal significant effects based on age, obesity, intervention duration, hypoxia duration, inspired oxygen fraction, and intervention type. Sensitivity analysis did not reveal any differences in the pooled results after a removing single study.

## Analysis of maximal power output

Six experiments involving 184 participants reported information on power output. Moderate certainty evidence (Fig. 3C) found that the two interventions have the same effect on maximal power output (SMD = 0.29, 95% CI [−0.17–0.76]; moderate to high heterogeneity), the certainty of evidence was downgraded due to high heterogeneity. Subgroup analysis (Table 2) did not identify significant effects based on age, obesity, intervention duration, hypoxia duration, inspired oxygen fraction, and intervention type. Sensitivity analysis did not reveal any differences in the pooled results after removing a single study.

**Table 2  Results of subgroup analysis and corresponding qualities of evidence (GRADE).**

| Study characteristics | Functional outcomes | | Muscle strength | | Power output | | $VO_{2max}$ | | $VO_{2peak}$ | | Exercise workload | |
|---|---|---|---|---|---|---|---|---|---|---|---|---|
| | N | Effect Size (95% CI) | N | Effect Size (95% CI) | N | Effect Size (95% CI) | N | Effect Size (95% CI) | N | Effect Size (95% CI) | N | Effect Size (95% CI) |
| **Age** | | | | | | | | | | | | |
| middle-aged | – | – | – | – | 4 | 0.35 (−0.40, 1.10)/L[a] | 4 | 0.90 (−0.68, 2.48)/VL[a,c] | 5 | 0.37 (0.03, 0.70)[*]/ H | 4 | −10.07 (−34.95, 14.80)/VL |
| older | 3 | −0.21 (−0.66, 0.24)/M[b] | 3 | −0.19 (−0.63, 0.26)/M[a] | 2 | 0.26 (−0.21, 0.74)/H | – | – | – | – | – | – |
| **Obese** | | | | | | | | | | | | |
| yes | – | – | – | – | 3 | 0.35 (−0.09, 0.80)/L[a] | 4 | 0.90 (−0.68, 2.48)/VL[a,c] | 4 | 0.41 (0.04, 0.79)[*]/M[a] | 3 | −13.83 (−48.95, 21.30)/VL[a,c] |
| no | 2 | −0.13 (−0.67, 0.41)/M[b] | 2 | −0.39 (−0.93, 0.14)/M[a] | 3 | 0.15 (−0.24, 0.55)/H | – | – | 2 | 0.13 (−0.35, 0.62)/H | – | – |
| **Intervention Duration** | | | | | | | | | | | | |
| ≤8 weeks | – | – | 2 | −0.39 (−0.93, 0.14)/M[a] | 4 | 0.23 (−0.15, 0.61)/M[a] | 5 | 0.39 (−1.12, 1.90)/VL[a,c] | 5 | 0.37 (0.04, 0.70)[*]/ H | 4 | −10.07 (−34.95, 14.80)/VL[a,c] |
| >8 weeks | 2 | −0.09 (−0.26, 0.08)/H | – | – | 2 | 0.26 (−0.21, 0.74)/H | – | – | – | – | – | – |
| **Hypoxia duration** | | | | | | | | | | | | |
| ≥60 min | 2 | −0.28 (−0.79, 0.24)/M[b] | 3 | −0.19 (−0.63, 0.26)/M[a] | 4 | 0.14 (−0.20, 0.48)/H | 3 | 0.52 (−1.43, 2.48)/VL[a,c] | 3 | 0.12 (−0.28, 0.52)/H | 2 | −26.86 (−65.06, 11.34)/VL[a,c] |
| <60 min | – | – | – | – | 2 | 0.78 (−1.01, 2.58)/L[a] | – | – | 3 | 0.55 (0.10, 0.99)[*]/ M[a] | 2 | 5.11 (−13.75, 23.98)/L[c] |
| **FiO2** | | | | | | | | | | | | |
| ≤15% | – | – | 2 | −0.51 (−2.76, 1.74)/L[a,c] | 4 | 0.35 (−0.40, 1.10)/M[a] | 5 | 0.39 (−1.12, 1.90)/VL[a,c] | 4 | 0.36 (−0.02, 0.74)/M | 3 | −16.89 (−44.14, 10.36)/VL[a,c] |
| >15% | – | – | – | – | – | – | – | – | – | – | – | – |
| **Intervention type** | | | | | | | | | | | | |
| Aerobic | – | – | – | – | 4 | 0.41 (−0.27, 1.10)/L[a] | 3 | 1.06 (−1.28, 3.40)/VL[a,c] | 4 | 0.39 (0.01, 0.77) [*]/M[a] | 3 | 0.03 (−14.73, 14.79)/L[c] |
| Resistance | – | – | 2 | −0.39 (−0.93, 0.14)/M[a] | – | – | – | – | – | – | – | – |

**Notes.**

N, no. of studies.

[*]$p < 0.05$. Hypoxia conditioning has greater effect than equivalent training in normoxia.

H, High-quality evidence; M, Moderate-quality evidence; L, Low-quality evidence; VL, Very Low-quality evidence.

Downgrade one level with reasons: a: Existence of high heterogeneity (Inconsistency); b: Differences in outcomes measures (Indirectness); c: Wide 95% CI and small sample size (Imprecision).

(a)

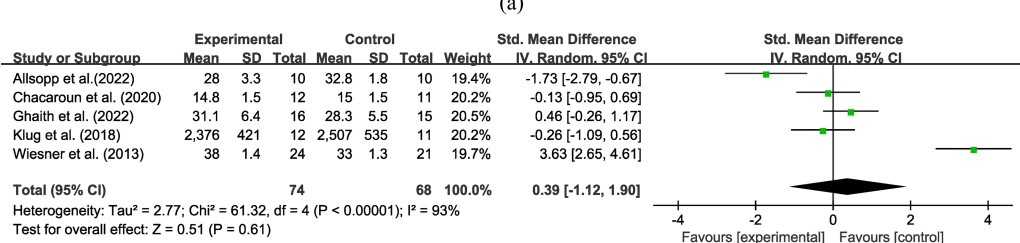

(b)

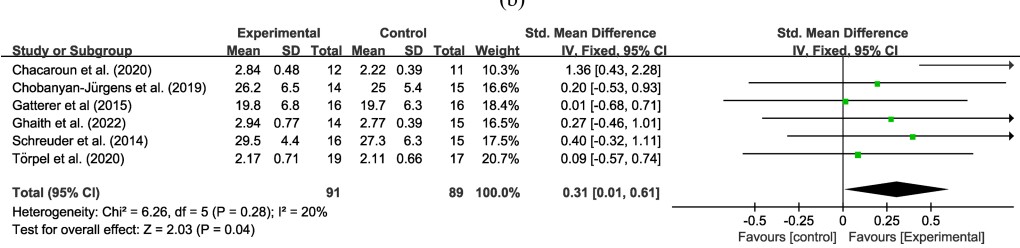

**Figure 4** Forest plot of the meta-analysis on (A) $VO_2max$ and (b) $VO_{2peak}$.

## Analysis of $VO_2max$

Five experiments involving 142 participants reported an outcome on $VO_2max$. Very low certainty evidence (Fig. 4A) found that the combined hypoxic conditioning and exercise strategy had a similar effect on $VO_2max$ as exercise alone (SMD = −0.39, 95% CI [−1.12–1.90]; moderate to high heterogeneity), the certainty of evidence was downgraded due to high heterogeneity and wide 95% CI. Subgroup analysis (Table 2) did not reveal significant effects on types of age, obesity, intervention duration, hypoxia duration, inspired oxygen fraction, and intervention type. Sensitivity analysis did not reveal any differences in the pooled results after removing a single study.

## Analysis of $VO_2peak$

Six experiments with 180 participants reported an outcome on $VO_2peak$. High certainty evidence (Fig. 4B) revealed that exercise in hypoxia led to a greater improvement in $VO_2peak$ than equivalent training in normoxia (SMD = 0.31, 95% CI [0.01–0.61]; low heterogeneity). In subgroup analysis (Table 2), exercise in hypoxia was a better exercise method in the middle-age group (SMD = 0.37, 95% CI [0.03–0.70], high certainty evidence), obese group (SMD = 0.41, 95% CI [0.04–0.79], moderate certainty evidence), intervention duration less or equal to 8 weeks (SMD = 0.37, 95% CI [0.03–0.70] high certainty evidence), hypoxia duration less than 60 min (SMD = 0.55, 95% CI [0.10–0.99] moderate certainty evidence), and aerobic exercise (SMD = 0.39, 95% CI [0.01–0.77], moderate certainty evidence). However, during the sensitivity analysis, the pooled estimate changed when the study by *Chacaroun et al. (2020)* was excluded, resulting in an equal effect between exercise in hypoxia and only-exercise groups (SMD = 0.19, 95% CI [−0.13–0.50]).

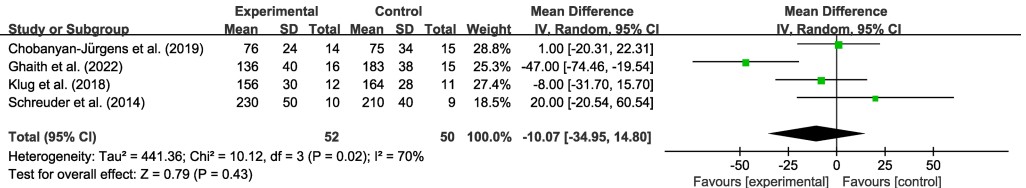

**Figure 5   Forest plot of the meta-analysis on exercise workload.**

## Analysis of exercise workload

Four studies involving 102 participants reported an outcome on workload. Very low certainty evidence (Fig. 5) found that the combined hypoxic conditioning and exercise strategy had a similar effect on workload as exercise alone (MD = −10.07, 95% CI [−34.95–14.80]; moderate to high heterogeneity), the certainty of evidence was downgraded due to high heterogeneity and wide 95% CI. Subgroup analysis (Table 2) did not show any significant effects based on age, obesity, intervention duration, hypoxia duration, inspired oxygen fraction, and intervention type. Sensitivity analysis did not reveal any differences in the pooled results after removing single study.

## The certainty of evidence in GRADE

Since the included studies are RCTs, the initial certainty of evidence was rated as high. The certainty of evidence for the effect of hypoxia conditioning on physical fitness ranges from very low to high, with the evidence levels for the subgroup analyses detailed in Table 2.

## DISCUSSION

The concern for the health of middle-aged and older population has become a primary focus of public health (*Concha-Cisternas et al., 2023*; *Edwards, 2012*; *Gell et al., 2023*; *McNicoll, 2002*; *Shirazi et al., 2023*; *Sillanpää et al., 2014*). In this context, we evaluated a strategy for improving physical fitness through the application of hypoxia as an intervention outside of exercise training. High certainty evidence revealed that training under hypoxic conditions led to a greater improvement in $VO_2$peak compared to training under normoxic conditions, while it had consistent effects on functional outcomes, muscle strength, power output, and $VO_2$max in middle-aged and older individuals as equivalent training in normoxia.

The meta-analysis found that the use of hypoxia conditioning resulted in significantly greater benefits for $VO_2$peak. Oxygen uptake capacity, a key indicator of cardiorespiratory fitness and aerobic capacity (*Caspersen, Powell & Christenson, 1985*; *Myers et al., 2002*; *Sugie et al., 2018*), was found to have marked improvement with hypoxic conditioning compared to normoxic conditions. One research has found that middle-aged and older individuals who undergo 8 weeks of HC experience a more significant improvement in $VO_2$peak compared to subjects who exercise under normoxic conditions. This may be attributed to the higher exercise intensity used in the study compared to other studies with similar intervention durations and training methods (*Chacaroun et al., 2020*). However,

contrasting outcomes have been observed in other studies. For instance, after an eight-week intervention combining hypoxia and HIIT in adults under 65 years old, no significant difference in $VO_2peak$ was found between the two groups. One possible explanation is the 20% reduction in training workload during HIIT under hypoxic conditions compared to normoxic conditions (*Ghaith et al., 2022*). In middle-aged individuals, eight weeks of HC resulted in a 10% increase in $VO_2peak$ (*Schreuder et al., 2014*). While hypoxic conditioning effectively improved the participants' physical fitness, no intergroup differences were observed. This might be due to the relatively low exercise intensity, the use of maximal heart rate as a reference for training intensity did not account for the fact that maximum heart rate is lower under hypoxia compared to normoxic ones (*Ozcelik & Kelestimur, 2004*; *Schreuder et al., 2014*). Our study supported the effectiveness of HC in enhancing oxygen uptake in middle-aged and older population. Notably, the observed superior effect of HC on $VO_2peak$ improvement primarily derives from a single included study, which underscores the necessity of rigorous validation through additional high-quality studies. $VO_2max$, determined through maximal exercise testing, represents the maximum rate of oxygen consumption during physical exertion, while $VO_2peak$ refers to the highest measured oxygen uptake attained during submaximal exercise testing and is commonly used to estimate $VO_2max$ (*Bassett Jr & Howley, 2000*; *Ryan, 2014*). Older adults frequently undergo submaximal exercise testing due to age-related challenges in sustaining maximal exertion levels during peak performance assessments (*Church et al., 2008*; *Schultz et al., 2020*). This physiological limitation in achieving maximal effort may explain why hypoxia conditioning demonstrate limited efficacy in improving $VO_2max$ within middle-aged and older populations. Only two studies reported information on both $VO_2max$ and $VO_2peak$, and future studies need more comprehensive data to compare the effects of hypoxia conditioning on aerobic capacity in middle-aged and older adults, and to explore potential associations between these two measures before and after the intervention.

For middle-aged and older population, hypoxia conditioning may not be the optimal choice for improving muscle strength. This finding aligns with the outcomes from a meta-analysis examining the effects of HC on muscle strength (*Csapo & Alegre, 2016*). While healthy young participants showed significant improvements in both absolute and relative 1RM after a seven-week resistance training regimen under hypoxic conditions compared to exercise alone, this effect may be attributed to the utilization of a lower volume, high-intensity training protocol, which is better suited for enhancing maximal strength (*Inness et al., 2016*). Such adaptive changes may also arise from the higher metabolic stress induced by hypoxia, triggering functional muscle adaptations (*Schoenfeld, 2013*; *Scott et al., 2017*). However, in healthy older individuals, an 8-week combination of hypoxia and resistance training did not yield significant changes in muscle strength compared to those exercising in normoxic conditions (*Allsopp et al., 2022*). This could stem from the delayed vasodilatory response of skeletal muscles due to aging and a sluggish upregulation of genes related to myogenic cell differentiation, collectively resulting in a delayed adaptation of older participants to resistance training in hypoxic conditions (*Casey et al., 2011*; *Gnimassou et al., 2018*).

In the context of aging-related conditions, about 2%–34% of people are affected by sarcopenia, resulting in diminished muscle strength and functionality (*Reijnierse et al., 2015*; *Shafiee et al., 2017*). A study involving 2,318 high-altitude residents over the age of 60 found that the incidence of sarcopenia was significantly higher than in those living at lower altitudes (*Ye et al., 2020*). After an 8-week stay in high-altitude areas, 24 healthy male subjects at 34.9 years showed that body weight was significantly decreased and association between lower oxygen saturation and loss of lean mass was established (*Wandrag et al., 2017*). Sarcopenia stems primarily from aging but is also influenced by external environmental factors. For instance, hypoxic conditions can lead to muscle atrophy and diminished contractile strength. This may be attributed to the impact of hypoxia, which not only accelerates the breakdown of skeletal muscle tissue, but also disrupts protein synthesis pathways (*Chen et al., 2014*). Therefore, hypoxia may be a risk factor for sarcopenia, and the application of hypoxic conditioning requires caution (*Ye et al., 2020*). Future research is needed to establish the causal relationship between hypoxia and sarcopenia in middle-aged and older populations.

The limited effectiveness of hypoxia conditioning in enhancing cardiorespiratory function and muscle strength may help explain why significant improvements in functional outcomes and maximal power output were not observed. Physical fitness such as muscle strength, cardiorespiratory capacity, and functionality typically decline with age (*Arnett et al., 2008*; *Sillanpää et al., 2014*; *Trombetti et al., 2016*). Although some studies suggest that hypoxic conditioning can improve physical fitness in middle-aged and older individuals, the evidence remains mixed. For instance, a 12-week intervention that combined hypoxia with aerobic and resistance trainings in older adults led to a notable increase in chair sit-to-stand repetitions compared to a control group engaged in exercise alone (*Park et al., 2019*), suggesting that hypoxia conditioning may enhance both aerobic and anaerobic capacities and muscle functionality more effectively than normoxic conditions. In middle-aged individuals, an 8-week hypoxia and aerobic exercises program resulted in a significant improvement in $PO_{max}$, potentially linked to the slight but non-significant increase in lean body mass (*Chacaroun et al., 2020*). However, our meta-analysis indicates that the effects of hypoxia conditioning combined with exercise on functional outcomes and power output are comparable to exercise alone. In older individuals, a 5-week exercise program that paired hypoxia with low to moderate level of resistance training did not yield additional benefits in muscle strength and $PO_{max}$, possibly due to the insufficient exercise load, although resistance training has been shown to increase $PO_{max}$ in younger cohorts (*Törpel, Peter & Schega, 2020*). Additionally, an 18-week intervention involving whole-body vibration under hypoxic conditions showed no significant improvements in the timed "Up and Go" test or lean body mass among older individuals compared to the baseline, underscoring the potential limitations of hypoxia conditioning in this age group and highlighting the need for further research to determine the most effective combinations of physical activities and hypoxic conditions (*Camacho-Cardenosa et al., 2020*).

The variability of individual responses to exercise interventions may also contribute to the suboptimal efficacy of HC interventions. For instance, although physical exercise demonstrates superior effects compared to usual hospital care in improving functional

outcomes, gait speed, and muscle strength among hospitalized older adults aged ≥75 years, at least 8.7% of individuals still exhibited no response to the intervention (non-responders) (*Sáez de Asteasu et al., 2019*). Evidence from insulin-resistant middle-aged women similarly reveals the presence of non-responders to both HIIT and resistance training (RT) interventions, with the prevalence of non-responders showing significant associations with exercise modalities (*Álvarez et al., 2017*). This phenomenon may originate from multiple factors including the difference in skeletal muscle microRNA expression, training status, psychological stress, and habitual physical activity (*Davidsen et al., 2010*; *Mann, Lamberts & Lambert, 2014*). Given the consistent emergence of non-responders across various exercise modes and the elevated prevalence of chronic diseases in older populations (*Guo et al., 2022*; *Izquierdo et al., 2021*; *Wang et al., 2020*), comprehensive exercise-related assessments, screenings, and individualized exercise prescriptions should be implemented prior to administering HC interventions to middle-aged and older adults.

The lower oxygen concentration, resulting in reduced actual training volume and exercise intensity, could explain the absence of significant intergroup differences. In middle-aged overweight individuals, a 4-week exercise program combining hypoxia and aerobic activities did not lead to significant changes in $VO_2max$. However, participants using HC experienced a significantly lower workload compared to the exercise-only group (*Wiesner et al., 2013*). In studies with lower training volume and intensity compared to normoxic training, it was revealed that even when power output and muscle strength were similar in both the HC group and the exercise-only group, HC may still serve as an effective alternative, particularly for populations unable to maintain a fixed training load, such as patients with osteoarthritis and middle-aged and older adults with lower physical fitness (*Ghaith et al., 2022*; *Schreuder et al., 2014*). Additionally, the influence of hypoxia on heart rate or exercise load should be taken into consideration, and conducting exercise experiments under hypoxic conditions may be more beneficial in determining the appropriate exercise intensity for participants undergoing HC.

A monitored, sustained low-oxygen supply combined with higher exercise intensity may be a crucial factor influencing the effectiveness of hypoxic conditioning. The study conducted by *Chacaroun et al. (2020)* introduced bias in the pooled analysis of $VO_2peak$, leading to significantly improved $VO_2peak$ in the subgroup analysis when hypoxia was combined with exercise compared to exercise alone. In comparison to other studies (*Camacho-Cardenosa et al., 2020*; *Gatterer et al., 2015*; *Schega et al., 2016*; *Schega et al., 2013*; *Törpel, Peter & Schega, 2020*), the research of *Chacaroun et al. (2020)* utilized sustained and stable low-oxygen supply (80% $SpO_2$) and higher exercise intensity (75% $HR_{max}$), which may have been more effective in inducing greater improvements in aerobic capacity than normoxic training.

Some limitations should be acknowledged. The limited number of studies included in this meta-analysis restricted our ability to perform subgroup analyses across various study characteristics. This constraint also impeded our ability to distinguish real asymmetry from chance, thereby precluding the creation of a funnel plot (*Sterne et al., 2011*). Additionally, the absence of detailed descriptions of the randomization processes in most included studies may introduce potential biases. Moreover, only four studies reported the mean

age of the final study population (*Chacaroun et al., 2020*; *Gatterer et al., 2015*; *Ghaith et al., 2022*; *Wiesner et al., 2013*), which could potentially introduce age-related biases by including individuals outside the intended middle-aged and older adult demographic. Although most high-quality research were published in English (*Morrison et al., 2012*), it may be necessary to consider material in a wider range of languages to broaden the scope of future research. While the observed heterogeneity was low, sensitivity analyses revealed reduced robustness in the pooled meta-analytic results for VO$_2$peak. This finding not only underscores the need for cautious interpretation of the efficacy of HC on improving VO$_2$peak outcomes, but also emphasizes the necessity to incorporate and systematically analyze additional studies in future investigations.

The randomization procedures in 11 out of 13 RCTs were marked as "Not clear," potentially impacting the validity of the final results. For instance, the study by *Chacaroun et al. (2020)*, which involved higher training intensity combined with a sustained, stable hypoxic supply, contributed to the poor robustness in the pooled analysis for power output. Similarly, the lack of robustness in the pooled analysis for VO$_2$max can be attributed to the younger age of the participants (*Wiesner et al., 2013*). Furthermore, the analysis of workload robustness was compromised by the relatively severe hypoxic levels used in another study (*Ghaith et al., 2022*), which resulted in a significant 20% reduction in absolute workload among participants undergoing hypoxia conditioning.

## CONCLUSIONS

Hypoxic conditioning has a greater effect on enhancing VO$_2$peak compared to equivalent normoxic training in the middle-aged and older population, while both have similar effects on maximal power output, functional outcomes, muscle strength, and workload. Future researches should explore different combinations of oxygen concentrations and exercise intensities in hypoxic conditioning to better understand its impact on middle-aged and older adults. Adopting more rigorous randomized controlled trial designs is also recommended to enhance the reliability of existing evidence.

## ACKNOWLEDGEMENTS

We appreciated Prof. Dr. Kirsten Legerlotz (ORCID: 0000-0001-6077-5711) for her constructive suggestions for our article.

### Funding

Fanji Qiu is supported by a grant from the China Scholarship Council (Grant No. 202106520004). The article processing charge was funded by the Open Access Publication Fund of Humboldt-Universität zu Berlin. The funders had no role in study design, data collection and analysis, decision to publish, or preparation of the manuscript.

## Grant Disclosures

The following grant information was disclosed by the authors:
China Scholarship Council: 202106520004.
Open Access Publication Fund of Humboldt-Universität zu Berlin.

## Competing Interests

The authors declare there are no competing interests.

## Author Contributions

- Fanji Qiu conceived and designed the experiments, performed the experiments, analyzed the data, prepared figures and/or tables, and approved the final draft.
- Jinfeng Li performed the experiments, analyzed the data, authored or reviewed drafts of the article, and approved the final draft.
- Liaoyan Gan analyzed the data, authored or reviewed drafts of the article, and approved the final draft.

## Data Availability

This is a systematic review/meta-analysis.

## Supplemental Information

Supplemental information for this article can be found online at http://dx.doi.org/10.7717/peerj.19348#supplemental-information.

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
