# Peer review of "Effect of hypoxia conditioning on physical fitness in middle-aged and older adults—a systematic review and meta-analysis"

_PeerJ, doi:10.7717/peerj.19348_

## Round 0.1 · original submission · Major Revisions

· Academic Editor

Major Revisions

Two reviewers give lots of feedback and comments. I consider these opinions to be valuable and constructive. Please read them carefully, address questions/comments individually, and revise your manuscript accordingly. To make your revised manuscript, please highlight each comment/question with colored text. And, the rebuttal letter should be clear and exact.

Reviewer 1 ·

Basic reporting

This article focuses on the impact of hypoxic training on the physical fitness of middle-aged and elderly populations, aiming to explore exercise strategies suitable for these groups. This aligns with the current need to improve the health of middle-aged and elderly individuals in the context of population aging.

Experimental design

The research employs a systematic review and meta-analysis, reported in accordance with PRISMA guidelines. The study design is clear, with a detailed literature screening process, including search strategies for databases such as PubMed and Embase, inclusion and exclusion criteria, and the use of Endnote for literature management, enhancing the reproducibility of the research. The data analysis methods are appropriate, and the results are presented clearly using forest plots, subgroup analyses, and sensitivity analyses to demonstrate the effects of hypoxic training on various physical fitness indicators.
However, there are still some unclear aspects of this article and suggestions for revisions and further discussion are needed.

Validity of the findings

1. The presentation of VO₂max and VO₂peak in the text should use the formal notation "VO₂max" and "VO₂peak." Both indicators reflect the cardiopulmonary capacity of the subjects, and it is recommended to explain their significance in the outcomes to clarify their roles.
2. The text lists "Chacaroun et al., 2020a" and "Chacaroun et al., 2020b," but the two articles listed in the references have identical titles and sources. Please verify this. Similarly, the two articles by "Csapo et al., 2016" in the references also have identical titles and sources; this should also be checked.
3. The meta-analysis results show high heterogeneity in some indicators (e.g., VO₂max, muscle strength, functional tests), but the article does not sufficiently discuss the potential sources of heterogeneity. It is recommended to explore possible sources of heterogeneity, such as differences in the number of middle-aged versus elderly participants, the proportion of obese individuals within these groups, hypoxic conditions (e.g., oxygen concentration, exposure duration, continuous or intermittent exposure), and variations in exercise type and intensity.
4. Lines 383–390 discuss the higher incidence of sarcopenia among high-altitude residents and its possible association with hypoxia, but this argument is based on a single study, which weakens the evidence. It is suggested that the potential causes of sarcopenia be first described, and then, its connection to hypoxia is elaborated on.
5. In lines 436–439, how is it determined that the hypoxic supply in this study is "sustained" and "stable"? Additionally, how does the reported hypoxic level of ~13% correlate with a SpO₂ of 80%?
6. In the subgroup comparisons (Table 2), the hypoxic conditions are grouped into ≤15% and >15%. What is the basis for using 15% as the cutoff for grouping? How does this affect the test results for the subjects? It is recommended to provide an explanation.
7. In the meta-analysis, the only result that reached statistical significance was VO₂peak. However, among the six studies compared for this indicator, only one study showed a significant difference (Figure 4b). Concluding that hypoxia has a "great effect" on VO₂peak based on this might lead to excessive bias.
8. If the goal is to explore efficient methods to improve the physical fitness of middle-aged and elderly individuals, even if hypoxic training shows significant differences, is it a practical, feasible, and easily implementable training method in real-world applications?

·

Basic reporting

Comments to the Author
I would like to thank the section editor for providing me the opportunity to review the manuscript entitled “Effect of hypoxia conditioning on physical fitness in middle-aged and older adults - a systematic review and meta-analysis” To my knowledge, no similar systematic review and meta-analysis has been done, so this paper has the potential to make a new contribution to the field. That said, I have offered some specific input below for the author/s to consider, which I hope they will find helpful.
I wish the author/s all the best with this manuscript.(The complete review comments are provided in the uploaded PDF file.)

Experimental design

Some methods need to be reconfirmed and revised.(The complete review comments are provided in the uploaded PDF file.)

Validity of the findings

Statistical results and discussions need to be reconfirmed, revised or supplemented.(The complete review comments are provided in the uploaded PDF file.)

---

## Round 0.2 · Minor Revisions

· Academic Editor

Minor Revisions

Two reviewers give several suggestions and comments on your manuscript. Please read these suggestions and comments carefully and revise your manuscript accordingly. I will make my decision after read your revision.

Reviewer 1 ·

Basic reporting

In the text, references are made to Chacaroun et al., in the 2020 literature (Chacaroun et al., 2020a, Chacaroun et al., 2020b). However, in Figure 2 and the various tables, only Chacaroun et al. (2020) is listed, making it impossible to distinguish which of the two references is being cited. It is recommended to specify which of the included references is being referred to.

Experimental design

no comment

Validity of the findings

no comment

Additional comments

L. 365 – 385, The discussion addresses the unclear randomization procedures, which is indeed a significant limitation. The authors correctly identify methodological issues in the studies by Chacaroun et al. (2020a) and Ghaith et al. (2022) that may affect result reliability. However, the discussion should more deeply analyze why hypoxic training is effective for VO₂peak but has limited effects in other studies.

This article provides valuable evidence supporting the positive impact of hypoxic conditioning on VO₂peak in middle-aged and older adults. However, there is room for improvement in the explanation of heterogeneity and future research directions.
I recommend strengthening those parts according to the limitations of limited articles and subgroup analysis.

·

Basic reporting

1. This study has a clear structure, complies with the PRISMA guidelines, and provides a PROSPERO registration number.
2. The background section provides sufficient information on the relevance of hypoxic training (HC) for middle-aged and elderly people.
3. The discussion section also cites appropriate literature to support the relevant arguments.
4. The charts are clear and well labeled, which helps to present the research results.

Experimental design

1. The inclusion and exclusion criteria are clear and reasonable.
2. The methodology met systematic review standards and provided a detailed search strategy.
3. While subgroup and sensitivity analyses were conducted, certain aspects require further
clarification.Please refer to <3. Validity of the findings> for issues that need to be corrected and clarified

Validity of the findings

Here are some suggested revisions:
1.Line357-359
According to the Cochrane Handbook (Higgins & Green, 2011), when fewer than 10 studies are included, the symmetry of the funnel plot may merely reflect random variation. Considering the limited number of studies on various physical fitness indicators in this research, it is recommended to exclude the publication bias analysis.
2.Several physical fitness indicators in this study exhibited low heterogeneity (I² < 50%) (e.g., functional outcomes, muscle strength, and VO₂ peak), yet subgroup analyses were still conducted. It is recommended that the authors clearly state the rationale for these subgroup analyses and justify their necessity.
3.Additionally, please clarify whether a sensitivity analysis should be performed first to ensure the robustness of the results before determining the need for subgroup analyses, aligning with the appropriate statistical framework. (Line 332-344- The authors indicated that the subgroup analysis for VO₂ peak had a significant impact on the results. However, in the sensitivity analysis, resulting in an equal effect between exercise in hypoxia and only-exercise groups after excluding the study by Chacaroun et al. (2020a) (SMD = 0.19, 95% CI: -0.13 to 0.50). This approach may affect the robustness of the findings. It is recommended to clarify whether a sensitivity analysis should be conducted first to ensure that the influence of included studies is controlled before proceeding with the subgroup analysis).
4. Table 2
Table 2 includes the subgroup analysis results for Intervention Type (aerobic exercise and resistance training); however, this variable is not mentioned in the main text (line 303-353). This discrepancy may lead to inconsistencies in the reporting of results. It is recommended to ensure consistency between the table and the text.

---

## Round 0.3 · accepted · Accept

· Academic Editor

Accept

The revised manuscript is satisfactory in general. However, it needs several grammar and spelling checks. Please read it again before resubmission.

Reviewer 1 ·

Basic reporting

there are still some grammar error or wording mistakes, and some APA format errors in context, such as L.47-50, [「Aging is a primary risk factor ..., which also "impairs" physical and ... ("Goodpaster et al. 2006; Niccoli & Partridge 2012; Partridge et al. 2018") .」 the comma should be added before the publication year.
Please recheck again to make sure the grammar and APA format are correct in context.

Experimental design

no

Validity of the findings

Although the main effect of the articles is only from a small number of articles, the results should be valid through the various steps of systematic review and integrated analysis, and the authors also discuss the methodology and results thoroughly.

Additional comments

no

·

Basic reporting

Abstract
1. Line 22-23
The sentence "This systematic review and meta-analysis of randomized controlled trials to investigate the efficacy..." contains a grammatical error. It is recommended to revise it as: "This systematic review and meta-analysis aims to investigate the efficacy..."
2. Line 34
The sentence "while very low to moderate certainty evidence shown that..." should be "while very low to moderate certainty evidence showed that..."

Experimental design

No comments.

Validity of the findings

No comments.